# Bacteria as Biocontrol Tool against Phytoparasitic Nematodes

**DOI:** 10.3390/plants10020389

**Published:** 2021-02-18

**Authors:** Varvara D. Migunova, Nicola Sasanelli

**Affiliations:** 1A.N. Severtsov Institute of Ecology and Evolution, Russian Academy of Sciences, Leninsky Prospect 33, 119071 Moscow, Russia; 2Institute for Sustainable Plant Protection, CNR, Via G. Amendola 122/D, 70126 Bari, Italy; nicola.sasanelli@ipsp.cnr.it

**Keywords:** biological control, *Bacillus*, *Pseudomonas*, *Serratia*, bioformulations

## Abstract

Phytoparasitic nematodes cause severe damage and yield losses to numerous agricultural crops. Considering the revision of the EU legislation on the use of pesticides on agricultural crops, control strategies with low environmental impact are required. The approach based on the use of bacteria seems particularly promising as it also helps to reduce the applied amounts of chemicals and stabilize ecological changes. This paper gives an overview of the main types of bacteria that can be used as biological control agents against plant parasitic nematodes and their interrelationships with plants and other organisms. Many experiments have given positive results of phytoparasitic nematode control by bacteria, showing possible prospects for their application. In vitro, greenhouse and field experiments have shown that bacteria can regulate the development of ecto- and endoparasitic nematodes by different modes of action. Triggering the induction of plant defense mechanisms by bacteria is seen as the optimum tool because the efficacy of bacterial treatment can be higher than that of chemical pesticides or at least close to it. Moreover, bacterial application produces additional positive effects on growth stimulation, raises yields and suppresses other pathogenic microorganisms. Commercial formulations, both as single bacterial strains and bacterial complexes, are examined.

## 1. Introduction

Plant parasitic nematodes (PPNs) are widespread in nature and they can cause severe damages and yield losses to numerous agricultural, forestry, ornamental and officinalis plants [1,2]. Annual worldwide losses caused by PPNs on the most economically important crops (vegetables, fruits, and nonedible field crops) range between 12 and 25% of the total production, for a total of over USD 80 billion [3,4]. Plant parasitic nematodes have also been found in association with bacterial and fungal pathogens which utilize PPNs penetration pathways in plants to reinforce plant disease symptoms [5,6,7,8]. To control these complex diseases and to improve the plant’s health it is necessary to limit PPNs in the soil and suppress soil borne plant pathogens.

The use of chemicals in agriculture is currently restricted owing to increased EU legislation stringency [9]. This stimulates the search for alternative, environmentally friendly, nematode control strategies, that could be useful in a sustainable and organic agriculture [10]. Among these alternatives the use of bio-pesticides is an important tool in the integrated pest management [11] including the use of microorganisms and their metabolites. Pesticides and growth regulators of microbial origin have proved their potential in sustainable agriculture development and consequently in the development of ecofriendly nematode control measures [12]. Considering that microbial communities suppress nematode populations and determine soil suppressiveness [13,14], application of biocontrol agents, natural inhabitants of an ecosystem, can allow the reduction of the use of synthetic pesticides and stabilize ecological changes.

Plants and bacteria exist in intimate association within agricultural ecosystems. Bacteria may be attached to the root surface or phylloshere, forming symbiotic relationships with plants or existing as free-living organisms in soils [15].

According to Glick, soil bacteria, which facilitate plant growth and are found in association with plant roots and sometimes found on plant leaves or flowers or within plant tissues, are often termed plant growth-promoting bacteria (PGPB). Treating plants with PGPB, it is possible to observe: (i) biomass increase; (ii) nitrogen, phosphorus and iron content increase; (iii) increase in the length of roots and shoots; (iv) enhance seed germination; (v) greater resistance to disease and to various environmental stresses; (vi) increased production of useful secondary metabolites; and (vii) plant nutrition improvement [16].

Species of bacteria belonging to *Agrobacterium* sp., *Arthrobacter* sp., *Azotobacter* sp., *Clostridium* sp., *Desulfovibrio* sp., *Serratia* sp., *Burkholderia* sp., *Azospirillum* sp., *Bacillus* sp., *Chromobacterium* sp., and *Corynebacterium* sp. were reported for management of nematodes [17].

The suppression of PPNs by PGPB is achieved through different mechanisms based on the capacity of microbes to compete effectively for ecological niche, colonize plant surface and produce nematicidal and antimicrobial compounds (antibiotics, toxins, siderophores, hydrolytic enzymes, etc.).

Bacteria and their metabolites affect both plant and microbial community [18,19]. Direct antagonistic effect can be achieved by parasitism, antibiosis, or competition for nutrients or infection sites. Indirectly, bacteria can enhance host defense mechanisms provoking induced systemic resistance (ISR) [20].

Particular attention is paid to the members of the families *Bacillaceae* and *Pseudomonadaceae,* focusing on their applications to control the nematode of the *Meloidogyne* genus. Bacilli and pseudomonads widely occur in natural environments, especially in the root system of plants [21,22].

Biological control potential of PGPB against the two PPNs *M. incognita* and *Heterodera glycines* and the use of PGPB to prevent nematode damage to plants have been recently reviewed, emphasizing the importance of the application of bacteria for the management of these two PPNs and the mode of actions of PGPB [23,24]. These reviews focus on the mechanisms at the base of nematode suppression by PGPB (Volatile Organic Compounds, lytic enzymes, hydrogen cyanide, induced systemic resistance, lower ethylene, etc.) and on commercial *Bacillus* products.

In the present review, the authors highlight the interrelationships among plants, nematodes and pathogenic microorganisms, with particular focus on roots, stems and leaves. Ecto- and endoparasitic nematodes of roots, stems and leaves were selected. Growth stimulation and antagonistic activity of PGPB under nematode infestation are also analysed. In this context, the existing commercial formulations and their efficacy are considered.

## 2. Bacteria as Biological Control Agents

Bacteria and PPNs as components of agroecosystems coexist in the environment consisting of soil and plants. The whole entirety of interspecies and interpopulation relationships occurs in agrosystems, which have the self-regulation ability based on feedback control. This co-regulation stabilizes the population numbers of plants, PPNs and PPNs regulators. Population numbers of PPNs depend on the immune-genetic characteristics of the host plant and the composition and traits of biological regulators of nematodes [25]. Intensification of agriculture leads to crop rotation and decreases in biodiversity of natural regulators of PPNs [19].

By the feeding type, PPNs can be classified into ectoparasites, semiendoparasites, migratory endoparasites and sedentary endoparasites [26]. Several species of PPNs that feed on roots (root knot, cyst, reniform, lesion, dagger nematodes), stems and leaves (*Ditylenchus*, *Aphelenchoides, Bursaphelenchus*) are widespread in nature and represent the most harmful parasites of plants [4,27]. The effect of bacteria on PPNs population density is particularly interesting in the cases of genera *Bacillus*, *Pseudomonas* and *Serratia* which have shown the highest efficacy as PGPB in their biological control for the last two decades [20,28,29].

The nematode attack on a plant passes through three stages: (i) beginning (asymptomatic), (ii) visible symptoms (yellowing, wilting, drafting, yield losses, etc.) and (iii) the final stage of the parasitosis (often ending in the plant death depending on PPNs density). At each stage, changes occur both in the plant and in the interactions between the plant, the PPNs and biological control agents.

Biological control of PPNs by bacteria at different stages of the life cycle of nematodes is verified by in vitro tests, pot experiments and field trials.

### 2.1. In Vitro Tests

Nematicidal activity against the second stage larvae (J2) of *Meloidogyne javanica* and *Heterodera filipjevi* was demonstrated in vitro by cultural filtrates of *Bacillus subtilis* OKB105 (100%) and *Bacillus cereus* 09B18 (83%) [30,31]. *Serratia proteamaculans* Sneb 851 demonstrated high nematicidal potential against *Meloidogyne incognita* with 99% and 61% mortalities for J2 and eggs, respectively [32]. The screening of 662 bacterial strains for nematicidal activity against *M. incognita* revealed that *Bacillus* was the major genus that demonstrated high nematicidal activity compared with the other genera. A percentage of 34% of the tested *Bacillus* strains caused *M. incognita* J2 mortality similar to that obtained by Aldicarb [33].

From in vitro tests on migratory PPNs, the endophytic bacterium K6, isolated from leaves of coffee plants, caused 65% mortality of *Pratylenchus coffeae* [34]. *Bacillus subtilis* OKB105 and *Bacillus amyloliquefaciens* B3 demonstrated their nematicidal activity against the nematodes of aerial parts of plants *Aphelenchoides besseyi*, *Ditylenchus destructor*, *Bursaphelenchus xylophilus* with mortalities of 85%, 79% and 100%, respectively [30]. In vitro mortalities of the dagger nematode *Xiphinema index* of 85%, 94%, 80%, 65% and 52% were observed by bacterial filtrate of *Pseudomonas chlororaphis*, *Pseudomonas fluorescens* FP805PU, *Serratia plymuthica*, *Bacillus cereus* and *Bacillus amyloliquefaciens*, respectively [35,36].

Efficacy of cell-free filtrate and bacterial cultures against PPNs may differ significantly. *Radopholus similis* mortality of 99 and 96% was observed in laboratory conditions by cell-free water extracts of two *Bacillus firmus* formulations (Bf-125 and Bf-106), respectively, whereas on the same nematode a 41% mortality was recorded in a non-sterile sand treated with different cell concentrations of *Bacillus firmus.* These different mortality percentages can be attributed to the competition with other microorganisms in soil ecosystem which results in a different efficacy between filtrates and cultures [37].

### 2.2. Pot Experiments

There are many tests demonstrating significant reduction of disease development and nematode reproduction by PGPB in plants [18,32,38,39,40,41,42,43].

Four bacterial agents were tested in a pot experiment in a screen house. Pots were filled with solarized mixture of sandy loam soil. In the pots sugar beet plants were treated with bacterial suspensions at 3 × 10^8^ CFU/mL concentration added into four holes made around the base of each sugar beet plant. *Bacillus pumilis, B. megaterium, B. subtilis* and *P. fluorescens* significantly reduced numbers of galls and egg-masses of *M. incognita* on sugar beet roots. The reduction was 73%, 69%, 71% and 60% for gall numbers and 74%, 68%, 65% and 61% for egg masses, respectively [44].

Interesting results were achieved by bacterial complex treatments against root-knot nematodes (RKNs) and *Xiphinema index* in grapevines. In pot experiments with naturally infested soils exhibiting a high level of PPN infestation the liquid, powder and unformulated mixtures of rhizobacteria were tested. The concentrations of the bacterial strains were 10^6^,10^8^ and 10^9^ CFU/mL. For the treatment, the roots of plants were soaked in the bacterial mixtures suspensions. *Bacillus amyloliquefaciens* FR203A, *B. megaterium* FB133M, *B. thuringiensis* FS213P and FB833T, *B. weihenstephanensis* FB25M, *B. frigoritolerans* FB37BR and *Pseudomonas fluorescens* FP805PU in different mixtures significantly suppressed *Meloidogyne ethiopica* and *Xiphinema index* reproduction and disease development compared to the untreated controls. Some bacterial complexes exhibited effects similar to the organophosphate nematicide cadusafos [45].

According to the “cry-for-help hypothesis,” plants attract entomopathogenic nematodes [46,47]. Bacterial suspensions of symbiotic bacteria of entomopathogenic nematodes (*Photorhabdus luminescens, Xenorhabdus* sp., *X. szentirmaii*) significantly reduced *Meloidogyna hapla* number of galls (51–67%), egg masses (48–68%) and the reproduction factor (RF) (55–62%). Cell-free supernatant of *P. luminescens, Xenorhabdus* sp., *X. szentirmaii* decreased the number of galls (51–74%), egg masses (72–83%) and RF (62–72%) of *Nacobbus aberrans* [48]. It was also demonstrated that the supernatant of *Xenorhabdus bovienii* suppressed the population number of *A. besseyi* [49].

### 2.3. Field Trials

Field experiments were carried out in Tieling (China) on 24 m^2^ plots. Tomato seeds treated by bacterial fermentation broth at 10^8^ CFU/mL were sown in plastic pots and later at seedling stages transplanted in field. In these conditions a reduction of 56% of root gall inedx caused by *M. incognita* was observed by *P. fluorescens* Sneb 825 treatments in comparison to the untreated controls [32]. *Bacillus cereus* 09B18 showed 76% and 44% reduction of *Heterodera filipjevi* white females per plant in greenhouse and field conditions, respectively, which were not significantly different from those obtained by Avermectin treatments [31]. In two-years field experiment *Bacillus aryabhattai* Sneb517 also demonstrated suppressive effect on J2 number and cyst formation of *Heterodera glycines* on soybean of 4- and 3-fold, respectively, compared with the untreated control [50]. *Bacillus subtilis* provided effective control of root knot and lesion nematodes on sugarcane in field conditions reducing the final nematode population density compared to the controls. For root knot nematode control level ranged from 46% to 56%, which was 1.5 times higher than that for the chemical control with carbofuran. Control of *Pratylenchus* spp. by *B.subtilis* on sugarcane ranged between 38 and 49% whereas it was 30–35% in carbofuran control [51]. *Bacillus subtilis* also significantly reduced root knot nematode and root lesion nematode populations (54%) and disease development (64%) on common bean [52].

In the Camaguey province (Cuba) *Bacillus thuringiensis* var. *kurstaki* LBT 3 treatments allowed 87% reduction of *Radopholus similis* population on banana plants [53].

## 3. Growth Stimulation and Productivity of Plants

PGPB frequently demonstrate both nematicidal and plant growth stimulation activities. *Serratia plymuthica* M24T3 isolated from pinewood nematode (PWN) showed a high nematicidal activity (100%) towards PWN [54]. This strain demonstrated high plant colonization activity and growth stimulation [54]. On the contrary, some PGPB can significantly stimulate plant growth but do not control nematodes, as was revealed by Aballay et al. [35] for *B. brevis* and *Comamonas acidovorans*.

Many reports have shown that PGPB can promote plant growth and enhance yield from plants affected by nematodes under greenhouse, microplot and field conditions [6,22,43,55,56,57,58,59,60]. A microplot experiment was carried out in the National Agriculture Research Center (Tsukuba, Japan) on low-humic and *M.incognita* infested soil to ivestigate the nematicidal effects of *Pasteuria penetrans*. Tomato plants were irrigated by *Pasteuria* suspensions at 2.5 × 10^9^, 5 × 10^9^ and 5 × 10^10^ CFU/mL concentrations. It has been demonstrated that tomato fruit yield increased by 46% compared to untreated control by application of *P.penetrans* [61]. Youssef et al., revealed that *B. pumilis* and Micronema (bacterial complex which included *Serratia* sp., *Pseudomonas* sp., *Azotobacter* sp., *Bacillus circulans* and *B. thuringiensis*) increased the length of shoot, fresh and dry weight of shoot and sugar beet root (tuber) weight in greenhouse conditions [44]. The percentage increase of these parameters was 34%, 44%, 88%, 88% and 48%, 88%, 72%, 152% for *B. pumilis* and Micronema, respectively. Treatment of tomato plants affected by *M. incognita* with *B. subtilis* Sneb 815 and *P. fluorescens* Sneb 825 increased plant height and root length by 25% and 30%, respectively, in field conditions [32]. Xiang et al. [33,62] demonstrated that *Bacillus velezensis* Bve12 and *Bacillus weihenstephanensis* Bwe15 increased early plant growth and enhanced cotton yield of plants grown in field conditions, affected by *M. incognita*. *Bacillus altitudinis* Bal13 increased soybean yield of plants affected by *H*. *glycines*. In other field trail, soybean plants infested by *R*. *reniformis* and treated with *B*. *mojavensis* Bmo3 and *B. velezensis* Bve2 resulted in a yield not significantly different from that obtained by the use of the Abamectin (c.n. Avicta) in the chemical control [63,64].

## 4. Modes of Action of Bacteria on Nematodes

Antagonistic interactions between bacteria and PPNs can be developed by more than one mode of action. The effect of bacteria on PPNs can be direct and indirect. Direct modes of action are colonization, parasitism and antibiosis (production of lytic enzymes, antibiotics, toxins, VOC (volatile organic compounds)). Indirect mechanisms include ISR, food supply for bacterivorous organisms (protozoa, nematodes), production of siderophores, hormones, phosphate solubilization, nitrogen fixation, transformation of bacterial microbiome.

Preemptive colonization can become a decisive factor in the competition for space and nutrients as bacteria interact with PPNs and plant pathogens [65]. Here it should be pointed out that an increase in the PGPB population numbers in the soil stimulates soil animal activity. Soil animals feed on bacteria causing a dissemination of bacterial cells in the soil and the increase of nitrogen and/or carbon mineralization. Nitrogen excretion by bacterivorous nematodes can promote bacterial growth and increase root colonization by PGPB [66]. However, this assumption needs more detailed investigation.

*Pasteuria penetrans* demonstrates parasitic mode of action on nematodes. Its spores get attached to the nematode’s cuticle, penetrate the body by germination tube and form endorspores inside the body. *Pasteuria* endospore adhesion is dependent on the nematode age. The age of phytonematode cuticule is affected by root exudates [67].

It is known that there are 323 nematode species belonging to 116 genera on which member of *Pasteuria* genera parasitize [68,69,70]. These include economically important PPNs: root-knot nematodes, cyst nematodes, root lesion nematodes, burrowing nematodes, foliar nematodes. *Pasteuria* is highly specific parasite which has been studied as biological control agent since 1980 [71] (Table 1).

Antagonistic mode of interaction of PGPB may include production of nematicidal and antimicrobial compounds. Only a few studies have so far characterized these substances and so they remain largely unknown. Nematicidal compounds of rhizobacteria were recently reviewed by Castaneda-Alvarez and Aballay [74]. The members of the family *Bacillaceae* produce proteases, chitinases, collagenases, lipases and complexes of enzymes, which affect different stages of the life cycle of PPNs. The ability to produce chitinases was also registered for *Serratia marcescens.* Pseudomonads produce glucanases, cellulases and pectinases to control *M. incognita.* (Table 1).

Crystal toxins of *Bacillus thuringiensis* are responsible for nematicidal effect against a wide spectrum of nematodes. Their high toxicity against PPNs makes *B. thuringiensis* the leading biocontrol agent [29,81]. Recently, it has been discovered that crystal proteins of *B. thuringiensis* may enter in PPNs through the stylet (Table 1).

Enthomopathagenic bacteria from the genera *Photorhabdus* and *Xenorhabdus* release toxins and antibiotic substances suppressing endoparasitic nematodes, inhabitants of roots and aerial parts of plants (Table 1).

It has been reported that hydrogen cyanide liberated by pseudomonads and hydrogen sulphide liberated by bacilli showed nematicidal effect on root knot nematodes, cyst nematodes, burrowing nematodes and dagger nematodes. The effect of hydrogen cyanide is related to inhibition of mitochondrial cytochrome oxidase [74].

Bacilli produce lipopeptides that suppress *M. incognita*: surfactin, bacillomycin D, fengycins, iturins and bacteriocins. Fluorescent pseudomonads produce antibiotic DAPG (2,4-diacetylphloroglucinol) which decreases juvenile mobility and enhances egg hatch of the potato cyst nematode *Globodera rostochiensis* (Table 1). Microbial community interactions, signals and communications with plants as well as biofilm formation could be regulated by antibiotics in low concentrations [86]. It has been demonstrated that nearly 10% of the *B. amyloliquefaciens* FZB42 genome synthesize antimicrobial metabolites. However, these compounds are not the main mode of PPNs suppression, they can act as inducers of plant-mediated mechanisms defense [6,14,79].

It has been shown that *Meloidogyne* spp. suppresses host defense responses to survive inside plant root. One of the underlying possible mechanisms is the prevention of calcium ion flux through sequestration of free calcium [27]. *Bacillus subtilis, B. cereus, B. pasteurii, B. amyloliquefaciens, B. mycoides, B. pumilus, B. sphaericus, P. fluorescence, Rhizobium leguminosarum, P. putida, S. marcescens* can provoke ISR [87]. ISR against *Meloidogyne javanica* and *M*. *graminicola* by fluorescent pseudomonads has been well documented [75,76,77]. DAPG has been shown acting as an inducing agent [65,86,88]. Cyclic lipopeptides and VOCs are the key triggers of ISR by Gram-positive endospore forming bacteria. It has been revealed that *B. firmus* I-1582 provoke ISR dependent on the plant species [89].

Nascimento et al. revealed that *Pseudomonas putida* UW4, which has no nematicidal activity, significantly decreased symptoms of *B. xilophilus* (PWN) disease in *Pinus pinaster* and the number of living nematodes [90]. Application of *P. putida* increased shoot and root growth of 7% and 61%, respectively, while in the untreated control affected by PWN the same parameters decreased by 41% and 35%. The main mechanism of the disease suppression is the 1-aminocyclopropane-1-carboxylate (ACC) deaminase produced by *P. putida* UW4. Through ACC deaminase production, *P. putida* UW4 reduces the deleterious ethylene levels in pine seedlings induced by PWN invasion. According to Nascimento et al., “Reducing deleterious ethylene levels and directly promoting plant growth, *P. putida* UW4 can boost plant defense systems, thus helping pine seedlings to overcome some of the negative consequences of PWN infection.” [90].

Siderophores can also be involved in ISR [79,91]. In fact, it is known that *P.fluorescens* CHAO, *P.fluorescens* WCS 374, *P.fluorescens* GRP3 produce siderophores acting as elicitors of ISR against tobacco necrosis virus, *Fusarium* spp. and *Rhizoctonia* spp., respectively. *Pseudomonas aeruginosa* 7NSK2 produce two siderophores under the low iron condition: pyoverdine and pyochelin. Pyochelin plays a role in tomato protection against damping off caused by *Pythium splendens*.

## 5. Mechanisms Involved in Plant Growth Stimulation

Plants affected by PPNs are under stress caused by lack of water and nutrients. PGPB can compensate this deficiency by an increase in nutrient availability (nitrogen, phosphorus and others).

Direct promotion of plant growth by PGPB includes the production of auxins, cytokinins, gibberellins, ethylene, abscisic acid, salicylic acid, jasmonic acid, strigolactones, nitrogen fixation, phosphate and potassium solubilization and sequestration of iron by bacterial siderophores [87,92,93]. *Bacillus licheniformis* FMCH001 enhanced the plant water use efficiency by increasing the root biomass of maize. These results might be caused by the upregulation of antioxidative enzyme catalase under both well-watered and drought stress conditions [94]. 2,3-butanediol (VOC) produced by *Bacillus* is responsible for significant improvement in plant growth [95].

## 6. Suppression of Phytopathogenic Microorganisms by Bacteria

The role of microbiome species composition is very important for the development of PPNs in soil [13]. Opportunistic phytopathogenic microorganisms as *Botrytis cinerea*, *Rhizoctonia solani*, *Fusarium oxysporum, Pythium ultimum* weaken the plants affected by nematodes causing nematode-fungal complex diseases. Production of chitinases and proteases by *S. plymuthica* C48 and pseudomonads, antibiotics (iturin A, surfactin, zwittermicin A, bacillomycin D, fengycin) by bacilli, siderophores (pseudobactin, pseudomonine) and hydrogen cyanide by *Pseudomonas fluorescens* suppresses development of phytopathogenic fungi [17,91,93,96,97].

It is recognized that some PGPB can control both PPNs and phytopathogenic fungi. *Serratia plymuthica* and pseudomonads are well known antagonists of fungal pathogens *Rhizoctonia solani* and *Verticillium dahlia,* moreover they reduced gall formation and egg masses on roots of tomato plants affected by *M. incognita* [6]. Bacilli and pseudomonads trigger pathways of ISR, which protect plants against microbial pathogens and PPNs [17,79]. Therefore, PGPB can suppress both PPNs and plant pathogenic microorganisms simultaneously.

## 7. Available Commercial Formulations of Bacteria

One sixth of active biopesticides ingredients registered in the United States of America (USA) against mites, insects and nematodes is of microbial origin which highlights the importance of microorganisms for biological control. Most of bacterial nematicides are registered in USA and Brazil (Table 2). *Bacillus subtilis* and *B. amyloliquefaciens* are prevalent in the market of products used to promote plant growth and biological control [73]. In Europe *Bacillus firmus* I-1582 and *B. amyloliquefaciens* FZB42 are the two microbial biological-based nematicides approved for use against RKNs in vegetable crops.

The majority of existing commercial formulations are represented by a single strain. The main targets for these products are root knot and cyst nematodes. At present, data about the efficacy of commercial bacterial formulations are scarce and inconsistent. It particularly concerns the bio-formulations based on *Pasteuria.* Products based on *Bacillus, Pseudomonas* and *Serratia* are more promising, because they not only suppress PPNs, but also stimulate plant growth and control plant pathogenic microorganisms. The application of the formulations Poncho and VOTiVO (based on *B. firmus*) demonstrated biological efficiency similar to the pesticides Avicta and Aldicarb [33,64,104].

Application of a consortia that includes bacterial, fungal and nematode antagonists is one of the most promising methods for RKN-control [114]. There are also several formulations consisting of two or more bacterial components (BioYield, Biostart, Micronema, Equity, Ag-Blends and others) proposed for the control of the root knot nematodes (Table 2).

However, future research is required to analyze the effects of single and complex formulations against PPNs in different seasons and soil types and their capacity to control pathogenic microorganisms and stimulate plant growth.

## 8. Conclusions, Problems and Future Prospects

Disturbance of biological balance and subsequent loss of microbiological diversity in an agrosystem can severely aggravate plant damage caused by PPNs determining the importance of bacteria into the soil. Introduction of bacteria in agroecosystems is completely different from the application of synthetic pesticides. Such pesticides are used to significantly reduce PPNs populations, whereas bacterial products trigger a series of processes that promote self-regulation of the ecosystem. Results of these processes are direct suppression or decrease of nematode population density and phyto-pathogenic microorganisms, ISR plant stimulation and changes in plant biochemical processes. Chemical pesticides are the agents alien to agroecosystems, whereas bacteria are important components of ecosystems. It is not clear whether it is necessary to introduce single strains of bacteria or bacterial consortia in agroecosystem. Must they be autochthonic or there may exist a universal solution?

One of the main reasons for appropriate application of bacteria as biological control agents against PPNs is the knowledge about their ecology and complex interactions in rhizobiome. To identify the factors determining PPNs regulation by bacteria and their survival in ecosystems, we need observations in natural and anthropogenic ecosystems, laboratory, field experiments and the study of conditions conducive to plant disease development. In conditions of insufficient data on bacterial application and inconsistent application results, the need for developing conceptual framework to manage PPNs populations is extremely urgent.

The major part of nematode biocontrol research has been carried out on sedentary endoparasitic nematodes. It has demonstrated significant suppression of population number of root knot and cyst nematodes belonging to genera *Meloidogyne* and *Heterodera.* On the contrary, data about migratory parasitic nematodes feeding on the surface and inside roots and parasitic nematodes of aerial part of plants are scarce. The strategy of biological control of such nematodes needs further development and experimental analysis. New data are needed to elucidate the interactions between bacteria and PPNs of different feeding types. It is important to take into consideration the life cycle of nematodes and the different conditions of their development.

The implication of the reviewed material essentially boils down to the following:

—It has been proven that bacteria as biological agents can be effective to control PPNs;

—Lack of data about their ecology and interactions in rhizobiome; therefore, further studies are required;

—Previous research focused on sedentary endoparasitic nematodes; this review broadens the scope of study as it looks at PPNs of other feeding types.

—PGPB not only suppress PPNs but also stimulate plant growth and productivity.

—Bacteria–nematodes interactions are based on multiple modes of action involving plants and other microorganisms.

—The market for commercial bacterial formulations is growing; the most effective and popular ones are based on *Bacillus*, *Pseudomonas* and *Serratia*.

The growing need to increase agricultural production and reduce environmental contamination causes future development in this domain. The multiple highly regulated in situ modes of action used by bacteria make them a safe and sustainable tool against plant parasitic nematodes. The most progressive strategy is the management of the population numbers of PPNs in the different agroecosystems.

The use of bio-pesticides is certainly an eco-sustainable method to control parasites and pathogens in a sustainable agriculture. It is natural that they are more sensitive to external factors, i.e., rain, temperature, chemical and physical characteristics of soil, which may adversely impact their effectiveness. Yet in many cases they are as effective as synthetic pesticides; even when they are not, agricultural goods produced using these bio-pesticides are likely to enjoy considerable demand for health and ecological reasons.

The market of bio-pesticides is booming. Their technological characteristics, such as persistence, transport and storage possibilities and shelf life, significantly improve their prospects on the bio-pesticide market.

This work is a contribution to the knowledge of the current situation in the biopesticides sector where bacterial formulations of different types are developed and traded.

## Figures and Tables

**Table 1 plants-10-00389-t001:** Bacteria and their modes of action against phytoparasitic nematodes.

Nematicidal Compounds	Species of Bacteria	Target Phytoparasitic Nematodes	References
Direct influence on nematode			
**Hyperparasitism**			
	*Pasteuria pentrans*	*Aphelenchoides besseyi, Globodera rostochiensis, Meloidogyne incognita, M.arenaria, M. hapla, M. graminicola, Pratylenchus penetrans, Radopholus* *similis*	[68,70,71]
**Antibiosis**			
**Lytic enzymes**			
Proteases	*Brevibacillus laterosporus*	*Bursaphelenchus xylophilus,* *Heterodera* *glycines*	[70,72]
	*Bacillus megaterium*	*M. graminicola* (juveniles)	[73]
Chitinases	*Serratia marcescens*	*M. hapla* (eggs and juveniles)	[74]
	*B. subtilis, B. pumilus*	*Meloidogyne* spp.	[73]
Collagenases	*B. cereus*	*M. javanica* (juveniles)	[74]
Lipases	*Bacillus thuringiensis* FB833T, *B. amyloliquefaciens* FR203A, *B. thuringiensis* FS213P	*Xiphinema index*	[74]
Complex chitosanase, alkaline serine protease	*B. cereus*	*M. incognita,* (juveniles)	[72]
Glucanases, cellulases, pectinases	*Pseudomonas*	*M. incognita*	[74]
**Antibiotics**			
2,4-diacetylphloroglucinol (DAPG)	Fluorescent pseudomonads	*M. javanica,* *M. graminicola*	[75,76,77]
		*Globodera rostochiensis* (eggs and juveniles)	[78]
Cyclic lipopeptides: surfactin, fengycin, iturins, acteriocins, polyketides, bacteriocins.	*B. amyloliquefaciens* FZB42, *B. subtilis*	*M incognita*	[18,70,73,79]
**Toxins**			
Hydrogen cyanide	Pseudomonads	Burrowing nematodes; cyst nematodes; dagger nematodes root-knot nematodes (juveniles);	[74,80]
Cry6Aa2 protoxin	*B. thuringiensis*	*M. hapla* (egg hatch, juvenile motility)	[29]
Cry5B	*B. thuringiensis*	*Meloidogyne* spp.	[81]
Cry1Ea11	*B. thuringiensis*	*Bursaphelenchus xylophilus*	[82]
Toxin A and toxin B, antibiotics	*Photorhabdus luminescens,* *Xenorhabdus budapestensis,* *X. szentirmaii*	*Aphelenchoides besseyi,* *M. incognita,* *Nacobbus aberrans*	[49,83,84]
**VOC**			
alkanes, alkenes, alcohols, esters, ketones, terpenoids, and sulfur families	*Bacillus* spp.	*M. graminicola* (juveniles)	[85]
	*Arthrobacter nicotianae*	*M. incognita* (juveniles)	[23]
	*B. amyloliquefaciens* FZB42	*M.incognita*	[79]

**Table 2 plants-10-00389-t002:** List of commercially formulated bacterial biocontrol agents against plant parasitic nematodes.

Bacteria	Name/Producer	Target	Observations/Findings	Reference
*Pasteuria pentrans*	Econem—Syngenta	*Belonolaimus* *longicaudatus*	Ineffective	[98]
*P. penetrans*	Econem—Pasteuria Bioscience, USA	*Meloidogyne* *incognita,* *M. arenaria*	Effect depends on plant culture and growing conditions.	[99]
*P. penetrans*	Econem—Nematech, Japan.	*M. incognita*	Biological efficiency 89%. Increase in marketable yield of sweet potato. The results are similar to Dichloropren.	[100]
*P. nishizawae* Pn1	Clariva PN—Syngenta, Brazil	*Heterodera* *glycines*	Ineffective	[62]
*Pasteuria* sp. *Ph3*	Naviva ST—Syngenta	*Rotylenchulus reniformis*	Inhibition of nematodes in cotton, soybean, vegetables.	[80]
*P. usage* Bl1 *+ Pasteuria* sp. *Ph3*	NewPro—Syngenta	*B. longicaudatus*	Inhibition of lance and sting nematodes in turf.	[80]
*B. amyloliquefaciens* FZB42	RhizoVital—AbiTep GmbH, Berlin, Germany	*M. incognita*	Reduction in J2, enhanced of root weight	[18]
*Burkholderia* *cepacia*	Deny—Stine Microbial Products, Madison, WI	*M. incognita*	Suppression of root-knot nematode population on bell pepper. Biological efficiency 60%	[101,102]
*Bacillus subtilis*	Stanes Sting—Imported from T. Stanes and Company Limited, India, by Gaara company, Egypt	*M. arenaria*	Reduction of J2 both in soil and roots as well as root galling, egg masses, biological efficiency 50%. Enhanced potato yield.	[103]
*B*. *firmus* I-1582	Poncho—Votivo Crop Science, Raleigh, NC	*Meloidogyne* spp., *Heterodera glycines*	Biological efficiency against *Meloidogyne luci* over 50%. Triggering ISR. Degradation of *Meloidogyne* eggs, colonization of plant roots.	[43,62,89]
*B*. *firmus* I-1582	Poncho—Votivo Crop Science, Raleigh, NC	*R reniformis*	Nematode control and yield similar to Avicta (Abamectin)	[63,64]
*B*. *firmus* GB-126	VOTiVO—Bayer, Germany	*R reniformis*	Reduction in number of females, eggs, and juvenile life stages. Cotton yield similar to aldicarb.	[104]
*B*. *firmus*	BioNem-WP—BioSafe—AgroGreen, Israel	*M. incognita*	Field efficiency 75–84%. Enhanced shoot height (29–31%) and tomato weight (20–24%)	[56,105]
*B*. *firmus*	BioNemaGon—Agri-Life, India	*Meloidogyne* spp., *Heterodera* spp., *Helicotylenchus* spp.	Reduction in nematode population and root infestation by nematodes in vegetables and herbs	[80]
*B. megaterium*	Bio-Arc	*Meloidogyne* spp.,	Reduction in J2, egg masses, eggs and reproduction factor. Enhanced shoot weight.	[106,107]
*B. megaterium*	Bio-Arc	*Tylenchulus semipenetrans*	Biological efficiency 88%,89% on baladi orange and lime	[108]
*B. methylotrophicus*	Onix—Laboratorio de Bio Controle Farroupilha S.A., Brazil	*M. javanica*	Ineffective on tomato plants	[109]
*Pseudomonas fluorescens*	Sheathgua (Sudozone)— Agriland Biotech, India	*Meloidogyne* spp., Cyst nematodes	-	[70]
*Serratia marcescens*	Nemaless—Agricultural Research Centre, Egypt	*M. incognita*	Reduction in J2, egg masses, egg numbers and reproduction factor. Improvement of tomato growth criteria: fresh weight of shoots and roots, length of both systems.	[106]
*B*. *amyloliquefaciens* IN937a, *B*. *subtilis* GB03	BioYield—Gustafson LLC, USA	*M. incognita*	Significant reduction in nematode eggs, juveniles and galls on tomato. Enhanced root weight.	[18,110]
*B. licheniformis,* *B. subtilis*	Nemix C	*Meloidogyne* spp.	-	[33]
*B. licheniformis* FMCH001, *B. subtilis* FMCH002	Presense—FMC Química do Brasil Ltd.a., Brazil	Plant parasitic nematodes	-	[70]
*B*. *subtilis*, *B*. *licheniformis*, *B*. *megaterium*, *B*. *coagulans*, *P. fluorescens*, *Streptomyces* spp.,	Pathway Consortia—Pathway Holdings, USA	Plant parasitic nematodes	-	[111]
*B. chitinosporus,* *B. laterosporus,* *B. licheniformis*	BioStart— Bio-Cat, USA	*Meloidogyne* spp.	Inconsistent effect	[112,113]
*Serratia* spp., *Pseudomonas* spp., *Azotobacter* spp., *B. circulans,* *B. thuringiensis*	Micronema—Agricultural Research Centre, Egypt	*M. incognita*	Significant reduction in J2, galls and egg masses (97%, 80% and 88%). Enhanced growth parameters: length of shoots, fresh and dry weight of shoots and roots.	[44]
47 strains of bacilli	Equity—Naturize Biosciences LLC, Jacksonville, FL, USA	*M. incognita*	Significant reduction in nematode eggs, juveniles and galls on tomato. Enhanced root weight.	[18]
Rhizobacteria and microbial metabolites produced during anaerobic fermentation of a microbial community	Ag-Blend—Advanced Microbial Solutions LLC, Pilot Point, TX, USA	*M. incognita*	Reduction of gall numbers, enhanced root weight.	[18]

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
