# Peer review of "Bacteria as Biocontrol Tool against Phytoparasitic Nematodes"

_plants, 2021, doi:10.3390/plants10020389_

Round 1
Reviewer 1 Report
The authors of the present study aimed to review the use of bacteria as a tool for biocontrol of parasitic nematodes of mainly agricultural crops. For this purpose, a literature review was conducted summarizing various effects bacteria have at parasitic nematodes and other pathogens, which often co-parasites on plants after they are invaded by nematodes.
Even though the aimed research question is of general concern since more eco-friendly methods of pests control without using chemicals with broad negative impact on other aspects of the ecosystem are necessary for sustainable agriculture there are a lot of indispensable features this feature lacks.
Some general comments:
- The methodological part of this review is totally missing! Where did you search the information? Which terms did you use to find literature? Did you have quality criteria for the final validation of studies? How many papers were included? This information are obligatory to conduct a comprehensible literature review
- What is your idea behind this review? A simple report of various effects is less interesting, I would suggest to comparatively assess the effect of different bacteria strains or genera on parasitic nematodes for example or to rank the susceptibility of pathogens responses assessed in the reported studies.
- The idea behinds this review is “the role that bacteria may play in biological control of PPNs are outlined, looking into the interrelationships among plants, nematodes and microorganisms”, what is in my perspective quite broad. And in the manuscript the discussion to some of the topics is quite shallow, such as chapter and 5 and 6.
- The conclusion is very weak (since the idea behind too broad or not specified enough!). What do you want to suggest? I do not see any additional value and discussion on the information you provide in the manuscript, which connects the findings from the scientific literature in some meaningful way.
Therefore, I cannot recommend the publication of this manuscript. However, I encourage the authors to re-submit the manuscript after major modifications.
Author Response
Response to REVIEWER 1
The present review was aimed to delineate the most recent situation on the use of bacteria as tool for plant parasitic nematodes biocontrol on the main agricultural crops. The literature review was conducted summarizing various bacteria effects on phytoparasitic nematodes and other pathogens, which often co-parasite causing more severe symptoms and damages to plants.
About some general comments formulated by the reviewer 1, it is important to consider that:
- The investigation is based on a bibliographic research carried out on internet using keywords. Research has been carried out on databases such as Scopus, Web of Science and on the home pages of the main companies of organic products. More than 400 recent articles were taken into consideration. In the References were considered and reported 114 articles.
- The idea of this review is to illustrate a general framework of the most recent studies on the use of bacterial formulations in plant protection against phytoparasitic nematodes in a sustainable agriculture. We don’t agree with the idea of a comparative assessment of the effect of different bacteria strains or genera on different parasitic nematodes. Since the reviewed papers applied different materials and methodologies, it is impossible to comparatively assess the effect of different bacterial strains.
- We agree with the idea that behind this review is “the role that bacteria may play in biological control of PPNs are outlined, looking into the interrelationships among plants, nematodes and microorganisms”. On the contrary we don’t agree with the comments on paragraphs 4 and 5.
Paragraph 4 entitled “Mode of action of bacteria on nematodes” is well developed with the description of the different mode of action of different bacteria genera and species that we have found in literature. Moreover, the results are also well sumarised in Table 1 in which reports different nematicide compounds such as lytic enzymes, antibiotics, toxins and VOCs produced by specific bacteria and effective against different nematode groups.
For paragraph 5 entitled “Mechanisms involved in plant growth stimulation” it is clear that it is rather short because the focus of the article is the use of bacteria for plant protection rather than the aspect linked to the mechanisms of bacteria as plant growth promoter.
- We modified the text and the title (Conclusion, problems and future prospects) improving the text and adding succinctly formulated statements. This work aims to be a contribution to the knowledge of the current situation of the biopesticides sector based on bacterial formulations of different types. In the paragraph we have highlighted the future prospects for the use of bacterial bio-formulations considering that they have numerous advantages (especially low risks for operators, for food and for the environment) compared to the use of synthetic chemicals.
We warmly thank the reviewer 1 for his/her contribution to improve our article and to consider the manuscript relevant (4 stars) for “Is the work a significant contribution to the field?”, “Are there appropriate and adequate references to related and previous work?” and “Is the English used correct and readable?” and enough (3 stars) for “Is the work well organized and comprehensively described?” and “Is the work scientifically sound and not misleading?”
Thanks again and Best regards.
Reviewer 2 Report
Line 13: It is essential to add explanation for why bacteria were focused in this study.
Line 22-23: It is better to describe what was obtained after commercial formations are examined.
Line 34; [9] is not an appropriate reference here.
Line 37: “exametabolites” is a collect term?
Line 42: This is also true for plants and fungi. It is necessary to describe reasons why authors focused on bacteria, not fungi.
Line 60: pseudomonades -> pseudomonads
Line 62-65: Recently, the following two review papers are published. It is necessary to specify differences/originality of this manuscript.
The Use of Plant Growth-Promoting Bacteria to Prevent Nematode Damage to Plants Gamalero, Elisa; Glick, Bernard R.
BIOLOGY-BASEL 9, (2020)
Biological control potential of plant growth-promoting rhizobacteria suppression of Meloidogyne incognita on cotton and Heterodera glycines on soybean: A review
Xiang, Ni; Lawrence, Kathy S.; Donald, Patricia A.
JOURNAL OF PHYTOPATHOLOGY 166, 449-458 (2018)
Line 70-71: This sentence needs some concrete results or appropriate references.
Line 100-120: It is necessary to describe the reason or speculation why efficacy differed between filtrate and culture.
Line 106-108: It is necessary to describe the inoculation methods of bacteria in brief. For example, inoculation, seed coating or drenching, concentration, sterile soil or nonsterile soil.
Line 106: It is better to add the strain names.
Line 110-113: In this context, the authors need to compare the results obtained with single inoculations, NOT the untreated controls. And add information on the inoculation method.
Line 114: Cadusafos -> cadusafos, the commercial name is not necessary.
Line 122; Need more information on the meaning of 56%, (gall index or No. of egg masses or ?), conditions of the field (soil type, location), name of crop, etc.
Line 126, 129, 130: For bacteria, their strain names are essential.
Line 127: what? gall index or No. of egg masses or eggs, or
Line 132: Here, the authors describe field size. It is better to show the field size in other places.
Line 138: The authors mention “highest”. But it is not clear in comparison with what?
Line 144: Add the strain name and the inoculation method.
Line 145-147: greenhouse, microplot or field conditions?
Line 164-165: It is better to describe why soil animal activity is stimulated by PGPB.
Line 218: It is better to describe how siderophores can be involved in ISR.
Line 271-272; These are not conclusion of this study.
Line 277; I do not agree with “bacteria are their natural components”, in particular when commercially formulated biocontrol agents are used in a foreign country.
Line 280-301; I consider that these descriptions are also not conclusion of this study.
Reviewer 3 Report
This is a well prepared review article on "Bacteria as biocontrol tool against phytoparasitic nematodes". The approach based on the use of microorganisms seems promising as it also helps to reduce applied amounts of chemicals and stabilize ecological changes. This study gives an overview of the main types of bacteria that can be used as biological control agents against plant parasitic nematodes and their interrelationships with plants and other organisms. Several experiments have given positive results of phytoparasitic nematode control by bacteria, showing possible prospects for their application. In addition, in-vitro, greenhouse and field experiments have showed bacteria can regulate the development of ecto-and endoparasitic nematodes by different modes of action. Certainly, triggering the induction of plant defense mechanisms by bacteria seems the optimum tool because the efficacy of bacterial treatment can be higher than that of chemical pesticides or at least close to it. Moreover, bacterial application produces additional positive effects on growth stimulation, raises yields and suppresses other pathogenic microorganisms. Excellent review and research article which is recommended for publication in "Plants".
Author Response
Response to REVIEWER 3
We warmly thank the reviewer 3 for his/her comments:
“This is a well prepared review article on "Bacteria as biocontrol tool against phytoparasitic nematodes". The approach based on the use of microorganisms seems promising as it also helps to reduce applied amounts of chemicals and stabilize ecological changes. This study gives an overview of the main types of bacteria that can be used as biological control agents against plant parasitic nematodes and their interrelationships with plants and other organisms. Several experiments have given positive results of phytoparasitic nematode control by bacteria, showing possible prospects for their application. In addition, in-vitro, greenhouse and field experiments have showed bacteria can regulate the development of ecto-and endoparasitic nematodes by different modes of action. Certainly, triggering the induction of plant defense mechanisms by bacteria seems the optimum tool because the efficacy of bacterial treatment can be higher than that of chemical pesticides or at least close to it. Moreover, bacterial application produces additional positive effects on growth stimulation, raises yields and suppresses other pathogenic microorganisms. Excellent review and research article which is recommended for publication in "Plants".
So, we don’t have anything to add.
Best regards
Round 2
Reviewer 2 Report
I judge that most parts were properly amended and this manuscript is now ready for acceptance.